# Bioaccumulation of Blood Long-Chain Fatty Acids during Hemodialysis

**DOI:** 10.3390/metabo12030269

**Published:** 2022-03-21

**Authors:** Tong Liu, Inci Dogan, Michael Rothe, Jana Reichardt, Felix Knauf, Maik Gollasch, Friedrich C. Luft, Benjamin Gollasch

**Affiliations:** 1Experimental and Clinical Research Center (ECRC), a Joint Institution of the Charité Medical Faculty and Max Delbrück Center (MDC) for Molecular Medicine, 13125 Berlin, Germany; tong.liu@charite.de (T.L.); maik.gollasch@med.uni-greifswald.de (M.G.); friedrich.luft@charite.de (F.C.L.); 2Lipidomix, GmbH, Robert-Rössle-Straße 10, 13125 Berlin, Germany; inci.dogan@lipidomix.de (I.D.); michael.rothe@lipidomix.de (M.R.); 3Department of Nephrology and Medical Intensive Care, Charité—Universitätsmedizin Berlin, Augustenburger Platz 1, 13353 Berlin, Germany; jana.reichardt@charite.de (J.R.); felix.knauf@charite.de (F.K.); 4Department of Internal Medicine and Geriatrics, University Medicine Greifswald, 17475 Greifswald, Germany; 5Helios, Klinikum Berlin-Buch, Schwanebecker Chaussee 50, 13125 Berlin, Germany

**Keywords:** exercise, lipidomics, erythrocytes, fatty acids, chronic kidney disease, hemodialysis

## Abstract

Long-chain fatty acids (LCFAs) serve as energy sources, components of cell membranes, and precursors for signaling molecules. Uremia alters LCFA metabolism so that the risk of cardiovascular events in chronic kidney disease (CKD) is increased. End-stage renal disease (ESRD) patients undergoing dialysis are particularly affected and their hemodialysis (HD) treatment could influence blood LCFA bioaccumulation and transformation. We investigated blood LCFA in HD patients and studied LCFA profiles in vivo by analyzing arterio–venous (A–V) LFCA differences in upper limbs. We collected arterial and venous blood samples from 12 ESRD patients, before and after HD, and analyzed total LCFA levels in red blood cells (RBCs) and plasma by LC–MS/MS tandem mass spectrometry. We observed that differences in arterial and venous LFCA contents within RBCs (RBC LCFA A–V differences) were affected by HD treatment. Numerous saturated fatty acids (SFA), monounsaturated fatty acids (MUFA), and polyunsaturated fatty acids (PUFA) n-6 showed negative A–V differences, accumulated during peripheral tissue perfusion of the upper limbs, in RBCs before HD. HD reduced these differences. The omega-3 quotient in the erythrocyte membranes was not affected by HD in either arterial or venous blood. Our data demonstrate that A–V differences in fatty acids status of LCFA are present and active in mature erythrocytes and their bioaccumulation is sensitive to single HD treatment.

## 1. Introduction

Chronic kidney disease (CKD) is a major public health problem worldwide, with a prevalence of approximately 10–15%, and the incidence of CKD continues to rise [1,2]. End-stage renal disease (ESRD) is the final, permanent stage of CKD, necessitating renal replacement therapies, notably hemodialysis (HD). ESRD patients face excess risk of developing cardiovascular disease (CVD), accounting for approximately 48% of total mortality [3]. Obesity, diabetes, hypertension, and hypercholesterolemia do not fully explain, or even contradict, this phenomenon [4]. Furthermore, the HD treatment in and of itself is implicated in CVD progression. Thus, nontraditional putative CKD-related risk factors, particularly when related to HD, must be evaluated [5].

Long-chain fatty acids (LCFAs) are an essential source of energy and a major cell membrane component of cell membranes. They participate in signaling pathways, influencing cell membrane structure and fluidity, affecting receptor affinity and ion channels [6,7]. A low omega-3 quotient in red blood cells (RBCs) is an independent risk factor for cardiovascular disease [8]. Eicosapentaenoic acid (EPA, C20:5 n-3) and docosahexaenoic acid (DHA, C22:6 n-3) are the two primary sources of dietary omega-3 fatty acids (FAs) derived from marine oil (Figure 1, orange font). Dietary supplementation with omega-3 FAs is effective in reducing cardiovascular risk in healthy people and ESRD patients [9,10], whereas studies that employed the combination EPA/DHA to standard of care therapy failed to derive any clinical benefit [11]. Several trials that tested purified EPA (JELIS, REDUCE-IT, EVAPORATE) were associated with reduced cardiovascular risk and the regression of atheroma coronary plaques [11]. Dietary EPA reduced ischemic events across the broad range of baseline eGFR categories [12]. Lower than normal blood polyunsaturated fatty acids (PUFA) n-3 levels [13] and increased blood monounsaturated FAs (MUFA) levels [10] are linked to increased cardiovascular risk in ESRD patients. Saturated FAs (SFA) may increase mortality by promoting vascular calcification in ESRD patients [14]. Dietary-induced changes in the FA composition of human plasma, platelet, and erythrocyte lipids follow a similar time course of several weeks [15,16,17]. Desaturases and elongases are involved in PUFA biotransformation [18] (Figure 1).

Our previous study [19] revealed that HD treatment alters the status of FAs and their oxidative metabolites, such as oxylipins, in arterial blood of ESRD patients. However, how HD treatment affects in vivo FA bioaccumulation and/or biotransformation in peripheral tissues, specifically whether FAs are produced, degraded, or stored in the blood as they pass through the arteries into peripheral tissues and organs and then from the venous end of the capillaries back into the venous circulatory system, is unclear. The arterio–venous (A–V) oxygen (O_2_) difference is the difference in the blood O_2_ content between the arterial blood and the venous blood. This difference shows how much O_2_ is removed from the blood in capillaries as the blood circulates in the body. ESRD patients present, before HD, with high ammonia plasma levels in arterial blood with a significantly positive arterio–venous difference [20]. The A–V blood glucose gradient is related to the fact that the peripheral tissues, especially the muscles, either store or burn part of the traversing glucose [21]. Likewise, non-esterified long-chain FAs (LCFAs) can exhibit positive (decumulation, i.e., loss) or negative (accumulation) A–V differences depending on the patient’s health or physical status and tissues perfused [22]. For example, negative differences in arterial and venous FAs within RBCs (RBC A–V differences) after passage through the upper limbs would be consistent with the bioaccumulation of those FAs in RBCs. In analogy, we tested the hypothesis that A–V differences in blood LCFAs are present in vivo and sensitive to single HD treatment. We collected arterial and venous blood samples of the upper limbs from ESRD patients, before and after HD, and measured the difference in the blood LCFAs content between the arterial blood and the venous blood by LC–MS/MS tandem mass spectrometry.

## 2. Results

### 2.1. Clinical Characteristics

The clinical features of the ESRD patients are shown in Table 1. The patients were diagnosed with FSGS (focal segmental glomerulosclerosis) (six patients), ADPKD (autosomal dominant polycystic kidney disease) (one patient), IgA nephropathy (one patient), hypertensive nephropathy (one patient), renal amyloidosis (one patient), drug-induced kidney injury (one patient), and cystic kidneys (one patient). All patients experienced macroangiopathic complications, including cardiovascular and cerebrovascular events and peripheral arterial disease. Appendix A shows that our patients did not have manifest diabetes but had hyperlipidemia.

### 2.2. Effects of Hemodialysis on Individual LCFAs in Plasma

The effects of hemodialysis treatment on individual FAs and their A–V differences are shown in Table 2. With the exception of C22:0 and C24:0, there were no A–V differences in plasma FAs levels before (pre-HD) and after HD (post-HD). Consistently, we did not detect A–V differences in SFA, MUFA, and PUFA plasma levels pre-HD and post-HD (Table 3). The negative pre-HD A–V difference in plasma levels of C22:0 and C24:0 disappeared after HD (Table 2).

There were no A–V differences in the dietary n-3/n-6 ratio, omega-3 quotient (EPA + DHA)/AA, EPA/AA, DHA/AA, and EPA/DHA ratios pre-HD and post-HD (Table 4 and Appendix A). To see whether desaturase activities were affected by single HD treatment, we calculated the following ratios: C18:3 n-6/C18:2 n-6 ratio, representing ∆6D, C20:4 n-6/C20:3 n-6 ratio, representing ∆5D, C16:1 n-7/C16:0 and C18:1 n-9/C18:0 ratios, representing ∆9D and DHA/DPA peroxisome functions, respectively (Table 5). The results show that pre-HD or post-HD values were not statistically significant (Table 5 and Appendix A).

### 2.3. Effects of Hemodialysis on Individual LCFAs in RBCs

Interestingly, most of the FAs in the erythrocyte membrane exhibited negative A–V differences before HD. These included C16:0, C18:2 n-6, C18:3 n-3 alpha, C20:1 n-9, C20:3 n-6, C20:4 n-6, C22:0, C22:1 n-9, C22:5 n-6, and C24:0 (Table 2). Consistently, we detected negative A–V differences in SFA, MUFA, and PUFA n-6 in RBCs before HD (Table 3). HD reduced these differences (Table 3). The individual SFAs were C16:0, C22:0, and C24:0. The individual MUFAs were C20:1 n-9 and C22:1 n-9. The individual PUFAs were C18:2 n-6, C20:3 n-6, C20:4 n-6, and C22:5 n-6. Notably, the omega-3 and n-3/n-6 RBC quotients remained unchanged pre-HD and post-HD (Table 4 and Appendix A). Similarly, C18:3 n-6/C18:2 n-6, C20:4 n-6/C20:3 n-6, C16:1 n-7/C16:0 and C18:1 n-9/C18:0, and DHA/DPA ratios, representing ∆6D, ∆5D, ∆9D, and peroxisome functions, respectively, did not show differences pre-HD and post-HD (Table 5 and Appendix A).

## 3. Discussion

To our knowledge, our study is the first to study the A–V differences of blood FA content between the arterial blood and the venous blood. Venous blood was withdrawn from upper limbs. In this case, measurement of FA consumption by the specific organ system requires the determination of the FA content of venous blood draining from that organ [20,21,22,23]. We applied this approach to better understand bioaccumulation and biotransformation of FAs in vivo, particularly whether or not the peripheral tissues, especially the muscles in the upper limbs, either produce, store, or degrade part of the FAs that pass through them in response to dialysis treatment. We were particularly interested to understand how the FAs in RBCs and plasma are modified by hemodialysis treatment, which is known to cause oxidative stress, RBC-endothelial interactions, vascular damage, and inflammation. Our study identified negative A–V differences of a number of FAs in RBCs after passage through the upper limbs before dialysis treatment (pre-HD). This is consistent with the bioaccumulation of those FCAs in RBCs. These effects were caused by all four main LCFA modules in RBCs: SFAs [i.e., palmitic acid (PA) (C16:0), docosanoic acid (C22:0) and lignocerine acid (C24:0)], MUFAs [i.e., eicosenoic acid (C20:1 n-9), erucic acid (C22:1 n-9), ω-3 PUFA [alpha-linoleic acid (C18:3 n-3)], and ω-6 PUFAs [i.e., gamma-linoleic acid (C18:2 n-6), dihomo-γ-linoleic acid (C20:3 n-6), arachidonic acid (C20:4 n-6), docosapentaenoic acid omega 6 (C22:5 n-6)] (Figure 1, bold font) in RBCs. With the exception of the SFAs C22:0 and C24:0, these differences were not observed in plasma. We found that the A–V differences in RBC LCFAs were affected by HD treatment (post-HD). We conclude that A–V differences in the fatty acids status of SFA, MUFA, and PUFA n-6 are present and active in mature erythrocytes, and that this status is sensitive to single HD treatment. We have no evidence that these changes are caused by altered biotransformation (Figure 1, Table 4 and Table 5). We found that the omega-3 quotient in erythrocyte membranes is not affected by HD in either arterial or venous blood, which aids clinical diagnostics of cardiovascular disease risk in healthy individuals and ESRD patients [24,25]. Furthermore, we observed that dialysis does not affect the dietary n-6/n-3 ratio (n-3/n-6) in both arterial and venous blood.

Previous studies have shown that RBC PUFAs reflect the phospholipid PUFA composition of major organs and could be used to monitor FA distribution in individual organs, for example, the heart [26,27,28]. In our study, we detected significant A–V differences in LCFAs in RBCs, i.e., numerous SFAs, MUFAs, PUFA (C18:3) n-3, and PUFAs n-6, which were affected by HD treatment. This indicates that LCFAs are present and active in mature erythrocytes and their bioaccumulation in peripheral tissues, namely the muscles in the upper limbs, is affected by renal replacement therapy. Consistent with other studies (for review see [10]), we previously detected decreased RBC EPA (C20:5 n-3) levels in the arterial blood of CKD patients compared with control subjects [19]. These changes were paralleled by decreased RBC C18:3 n-6 and C20:3 n-6 levels and decreased plasma levels of C18:2 n-6, C20:3 n-6, C20:4 n-6, C20:5 n-3, and C22:5 n-6 [10]. Together, the results indicate that there is an altered profile of n-3/n-6 fatty acids in ESRD patients, which is affected by hemodialysis treatment. With regard to the metabolism of these FAs (Figure 1) and our new findings in the present study, we speculate that direct endothelial–erythrocyte interactions may have contributed to this effect, rather than the uptake of fatty acids from plasma.

Finally, we investigated whether desaturase and peroxisome functions could be altered by HD treatment. Our data argue against a role of altered biotransformation of LCFA (Figure 1) in either plasma or RBCs by dialysis. A recent study used the alpha-linoleic acid/dihomo-γ-linoleic acid (AA/DGLA) ratio as a measure of ∆5D activity and found that a high AA/DGLA ratio is an independent predictor of cardiovascular risk and all-cause mortality in HD patients [29]. ∆9D is a rate-limiting enzyme for the conversion of SFA to MUFA [30]. Cho et al. found that ∆9D may exert cytoprotective effects by inhibiting the lipotoxic effects of excessive SFA accumulation [31]. An opposite effect was seen in a study in diabetes, which discovered that the risk of developing type 2 diabetes was associated with enhanced ∆6D and ∆9D activity in the erythrocyte membrane [31]. Our study provides motivation and suggestions for future studies on desaturase and peroxisome functions in cardiovascular complications of ESRD.

Our study has several limitations. First, the sample size of our study is small, and numerous FA levels pre-HD and post-HD showed only trends, which were not statistically significant. Second, the patient’s diet was not strictly regulated, and different dietary habits may have led to individual differences in FAs. Finally, race, BMI, sex, and age might have had an impact on the results of our study.

## 4. Materials and Methods

### 4.1. Participants of the Study

The study was authorized by the Charité University Medicine’s ethical committee, and signed informed consent was acquired. The research was appropriately registered: (ClinicalTrials.gov (accessed on 28 February 2019), Identifier: NCT03857984). Nine men and three non-pregnant women over the age of 18 were recruited if they had a history of renal failure demanding thrice-weekly HD. Patients had to have a stable HD prescription in order to participate in the trial. They had to be dialyzing by a native fistula or Gore-Tex^®^ graft. Noncompliance with their dialysis prescription, anemia with hemoglobin (Hb) less than 8.0 g/dL, or an active infection were all exclusion criteria.

### 4.2. Assessment

All patients were treated while seated. Dialysis was performed on the subjects using a Polyflux 170H dialyzer (PAES membrane, Gambro, Lund, Sweden), with the ultrafiltration rate maintained constant throughout the HD sessions. All dialysis sessions had the same relevant dialysis treatment parameters (blood flow rate of 250 mL/min, dialysate flow rate of 500 mL/min, double needle puncture method, dialysis period of 4 h 15 min, and 37 °C dialysate temperature). Arterialized (shunt) blood samples were obtained on the fistula arm right before the start of dialysis (pre-HD) and at the end of dialysis (5–15 min before termination, post-HD). At the same time points, venous blood was taken on the ipsilateral limb through subcutaneous arm vein puncture to determine the arterio–venous (A–V) difference of the LCFAs. The A–V difference is noticeable since the peripheral tissues, particularly the muscles, generate, store, or decay a part of the LCFAs that circulate through them. All blood samples were collected via 4 °C precooled EDTA vacuum extraction tube systems. In a qualified clinical laboratory, glucose, lipoproteins, and triglycerides were measured with standardized techniques.

### 4.3. Plasma and RBC Membrane LCFAs Profile Analysis

The preparation and handling of samples, reference standards, and HPLC–MS measurements were performed as described elsewhere [32,33]. Plasma samples (200 µL), added with 300 µL of 10M sodium hydroxide (NaOH), were subjected to alkaline hydrolysis at 60 °C for 30 min. The sample pH was then adjusted to six using 300 µL 58% acetic acid. The prepared samples were then subjected to solid-phase extraction (SPE) using a Varian Bond Elut Certify II column. The extracted metabolites were evaluated by LC–MS/MS using an Agilent 6460 Triple Quad mass spectrometer (Agilent Technologies, Santa Clara, CA, USA) and an Agilent 1200 high-performance liquid chromatography (HPLC) system (degasser, binary pump, well-plate sampler, thermostatic column chamber). A Phenomenex Kinetex column (150 mm 2.1 mm, 2.6 m; Phenomenex, Aschaffenburg, Germany) was used in the HPLC system. All samples were analyzed for total plasma and total RBC LCFAs.

### 4.4. Plasma and RBC Membrane LCFAs Cluster

The LCFA cluster was composed of 21 FAs which represent major components of plasma and erythrocyte membrane lipids. There were four main modules: SFAs [e.g., palmitic acid (PA) (C16:0) and stearic acid (C18:0)], MUFAs [e.g., palmitoleic acid (C16:1 n-7) and oleic acid (C18:1 n-9)], ω-3 PUFAs [e.g., EPA (C20:5 n-3) and DHA (C22:6 n-3)], and ω-6 PUFAs [e.g., linolenic acid (LA) (C18:2 n-6), arachidonic acid (AA) (C20:4 n-6)] (Figure 1) [34]. Taking into account the different meanings represented by the proportions of different FAs, the following indices were calculated: dietary n-6/n-3 ratio (n-3/n-6) [35] and omega-3 quotient [(EPA + DHA)/total FAs] [36,37]. Meanwhile, the enzymatic index of the desaturases that mainly synthesize MUFA and PUFA was calculated by the product/precursor ratio of the FAs participated: delta-6-desaturase (∆6D C18:3 n-6/C18:2 n-6), delta-5-desaturase (∆5D C20:4 n-6/C20:3 n-6), delta-9-desaturase (∆9D C16:1 n-7/C16:0, C18:1 n-9/C18:0) and peroxisome function [DHA/DPA (docosapentaenoic acid)] [38] (Figure 1).

### 4.5. Statistical Analysis

Descriptive statistics were obtained, and variables were checked for skewness and kurtosis to ensure that they met the normal distribution assumptions. To check if the data were normally distributed, we utilized the Shapiro–Wilk test. Levene’s test was used to demonstrate variance homogeneity. To evaluate statistical significance, arterial vs. venous values were compared using the paired *t*-test or the paired Wilcoxon test. The statistical significance value was set at *p* < 0.05. All data are displayed as median and interquartile range (IQR) or standard deviation (SD). All statistical analyses were performed using SPSS Statistics software (IBM Corporation, Armonk, NY, USA).

## 5. Conclusions

Our data demonstrate that HD affects RBC status by decumulating numerous LCFAs during passage through peripheral tissues, at least in the upper limbs. These changes may contribute to the increased cardiovascular risk in ESRD patients. Interestingly, RBCs, EPA, and DHA levels are not affected by single HD treatment.

## Figures and Tables

**Figure 1 metabolites-12-00269-f001:**
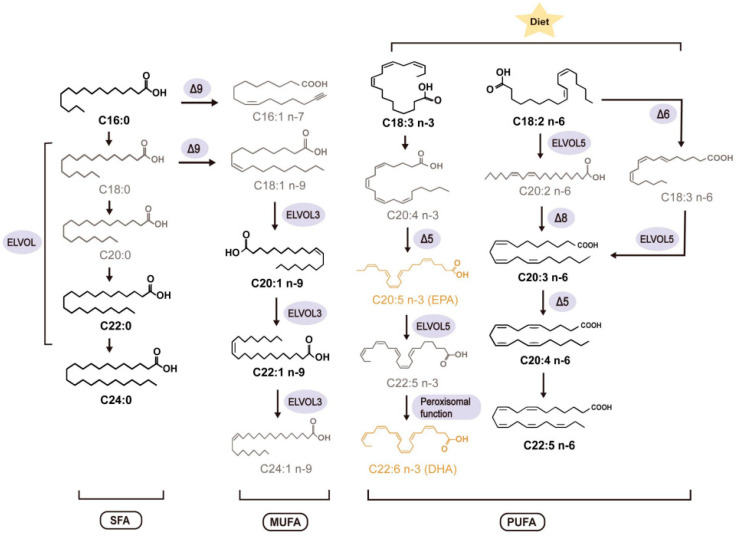
Fatty acid elongation and desaturation processes: SFA, saturated fatty acids; MUFA, monounsaturated fatty acid; PUFA, polyunsaturated fatty acids; ∆6D, delta-6-desaturase; ∆5D, delta-5-desaturase; ∆9D, delta-9-desaturase. Eicosapentaenoic acid (C20:5 n-3, EPA) and docosahexaenoic acid (C22:6 n-3; DHA) (omega-3 quotient) are highlighted in orange font. Bold font FAs showed negative A–V differences (bioaccumulation) in RBCs after passage through upper limbs before dialysis treatment.

**Table 1 metabolites-12-00269-t001:** Characteristics of patients (*n* = 12).

	Patients
Age (years)	72 ± 12
Sex	
Male (*n*)	9
Female (*n*)	3
Body mass index (kg/m^2^)	27 ± 3.3
Race (*n*)	Caucasian = 12
Cause of end-stage renal disease (ESRD)	
Focal segmental glomerulosclerosis (*n*)	6
IgA nephropathy (*n*)	1
Renal amyloidosis	1
Hypertension (*n*)	1
Drug induced (*n*)	1
ADPKD (*n*)	1
Cystic kidneys (*n*)	1
Complications	
Cardiovascular or Cerebrovascular (*n*)	12

Notes: Data are presented as mean + SD or frequencies. *n*, number.

**Table 2 metabolites-12-00269-t002:** Effects of hemodialysis on individual fatty acids in plasma and RBCs in the CKD patients before (pre-HD) and at cessation (post-HD) of hemodialysis (*n* = 12 each).

Fatty Acids	Chain	Pre-HD Median (IQR)				Post-HD Median (IQR)			
		Pre-HD Arterial	Pre-HD Venous	*p* Value, *t* Test (# Paired Wilcoxon test)	Pre-HD A–V Difference	Post-HD Arterial	Post-HD Venous	*p* Value, *t* Test (# Paired Wilcoxon Test)	Post-HD A–V Difference
**Total-Plasma (μg/mL)**									
Myristic Acid	C14:0	52.596 (34.455–74.978)	49.148 (35.486–61.274)	0.201	1.999 (−0.948–5.168)	69.936 (43.668–82.384)	55.294 (37.346–73.459)	0.05#	4.706 (2.599–11.296)
Myristolein acid	C14:1 n-5	3.234 (0.947–4.226)	3.043 (1.223–3.605)	0.328	0.054 (−0.289–0.518)	3.301 (1.639–5.112)	2.405 (1.255–4.273)	0.071#	0.114 (0.001–0.612)
Palmitic acid	C16:0	777.653 (573.133–955.245)	641.312 (546.790–1041.449)	0.259	38.589 (−7.998–83.389)	866.837 (608.336–1058.773)	757.319 (533.635–969.393)	0.081	81.436 (43.274–128.276)
Palmitoleic acid	C16:1 n-7	73.498 (49.472–95.766)	63.583 (50.81–93.068)	0.523	1.055 (−1.338–6.058)	66.312 (49.748–88.141)	56.138 (46.339–83.299)	0.16	5.162 (1.216–8.782)
Stearic acid	C18:0	127.090 (101.163–181.126)	133.632 (97.466–184.07)	0.92	6.2596 (−15.557–17.815)	153.798 (103.222–186.498)	127.594 (114.844–164.167)	0.222	11.989 (−6.372–44.691)
Oleic acid	C18:1 n-9	748.353 (613.3662–902.8834)	691.807 (586.015–991.742)	0.377	19.965 (−18.542–74.232)	825.9256 (638.758–986.897)	739.047 (587.658–929.974)	0.068	62.843 (38.444–85.592)
Linoleic acid	C18:2 n-6	562.592 (518.182–711.229)	581.052 (497.470–671.192)	0.397	9.3694 (−33.449–47.94)	630.349 (521.263–696.79)	590.649 (472.954–621.414)	0.05#	57.752 (24.256–71.841)
α-Linoleic acid	C18:3 n-3 alpha	13.945 (10.154–26.515)	15.015 (10.567–27.396)	0.719	−0.893 (−1.754–0.365)	20.302 (13.724–27.295)	16.611 (13.909–26.029)	0.347#	1.159 (−0.614–2.801)
γ-Linoleic acid	C18:3 n-6 gamma	8.567 (6.489–14.347)	9.140 (6.723–14.17)	0.54	0.788 (−0.981–1.764)	13.178 (6.819–14.602)	8.862 (6.362–12.27)	0.233#	1.352 (−0.135–2.425)
Eicosenoic acid	C20:1 n-9	5.34 (4.860–7.483)	5.787 (4.983–7.009)	0.583#	−0.119 (−0.364–0.242)	6.550 (5.726–7.261)	6.151 (5.337–7.128)	0.272#	0.424 (−0.126–0.769)
Eicosa-dienoic acid	C20:2 n-6	5.238 (4.487–6.676)	5.987 (4.542–6.569)	0.951	0.127 (−0.811–0.583)	5.0493 (4.126–6.253)	5.199 (4.435–5.981)	0.81	0.127 (−0.443–0.551)
Dihomo-γ-Linoleic acid	C20:3 n-6	44.416 (39.659–47.495)	41.626 (36.919–47.670)	0.929	1.638 (−1.572–2.888)	41.933 (39.131–47.516)	38.733 (35.309–43.548)	0.084	2.743 (0.195–7.246)
Arachidonic acid	C20:4 n-6	159.677 (128.831–169.561)	146.238 (136.060–158.549)	0.308#	5.4890 (−5.1079–24.766)	154.717 (135.01–179.618)	156.270 (132.641–164.080)	0.209#	11.712 (−1.005–24.280)
Eicosapenta-enoic acid	C20:5 n-3	17.969 (11.385–24.730)	15.828 (10.826–22.529)	0.168	0.744 (−0.584–2.694)	15.567 (12.080–23.746)	13.6374 (11.4568–20.6670)	0.2	0.903 (0.330–3.0154)
Docosanoate	C22:0	**0.436** **(0.243–0.803)**	**0.787** **(0.526–1.066)**	**0.012#**	**−0.359** **(−0.505–−0.146)**	0.5993 (0.4372–1.0188)	0.737 (0.253–0.853)	0.239#	0.213 (−0.192–0.346)
Erucic acid	C22:1 n-9	7.328 (6.898–8.415)	8.730 (7.581–9.637)	0.05#	−0.927 (−2.220–−0.385)	6.940 (6.582–7.573)	7.651 (7.203–8.111)	0.071#	−0.411 (−0.928–0.122)
Docosapentaenoic acid ω-3	C22:5 n-3	11.204 (8.975–13.927)	11.131 (8.707–13.017)	0.378	0.270 (−0.738–0.843)	11.945 (9.740–14.449)	11.412 (7.945–12.780)	0.267	0.782 (0.3697–1.1939)
Docosapentaenoic acid ω-6	C22:5 n-6	1.673 (1.401–2.283)	1.917 (1.604–2.483)	0.567	−0.034 (−0.214–0.094)	1.860 (1.471–2.430)	1.681 (1.504–2.419)	0.314	0.152 (−0.008–0.239)
Docosahexaenoic acid	C22:6 n-3	92.858 (74.276–116.778)	87.830 (81.820–99.411)	0.239#	2.672 (−3.136–10.778)	90.402 (83.206–119.631)	81.488 (76.371–105.830)	0.084#	5.538 (2.542–11.357)
Lignocerine acid	C24:0	**0.406** **(0.000–0.717)**	**1.685** **(0.855–2.110)**	**0.009#**	**−1.066** **(−1.910–0.285)**	0.350 (0.0003–0.594)	0.759 (0.043–0.993)	0.075#	−0.442 (−0.617–−0.042)
Nervonic acid	C24:1 n-9	2.918 (2.396–3.243)	3.070 (2.434–3.665)	0.568	−0.081 (−0.376–0.113)	2.732 (2.339–3.639)	2.701 (2.240–3.329)	0.497	0.190 (−0.460–0.450)
**Total-RBC (μg/g)**									
Myristic Acid	C14:0	22.836 (17.212–26.878)	26.270 (19.644–29.972)	0.084	−2.478 (−6.704–0.204)	26.098 (19.788–31.595)	22.349 (20.057–36.542)	0.465	0.295 (−4.188–2.787)
Myristolein acid	C14:1 n-5	0.127 (0.000–0.316)	0.040 (0.000–0.265)	0.889#	0.000 (−0.140–0.067)	0.301 (0.001–0.531)	0.138 (0.054–0.606)	0.182#	−0.060 (−0.198–0.002)
Palmitic acid	C16:0	**338.150** **(267.592–458.512)**	**400.180** **(294.720–538.391)**	**0.032**	**−40.619** **(−152.844–−0.799)**	367.433 (304.759–437.369)	343.501 (293.538–500.620)	0.308#	−5.556 (−78.396–30.194)
Palmitoleic acid	C16:1 n-7	16.349 (12.688–20.673)	20.828 (14.827–21.795)	0.091	−2.966 (−6.931–0.113)	17.388 (14.054–21.253)	17.995 (13.766–28.056)	0.083	−0.955 (−8.519–0.512)
Stearic acid	C18:0	244.115 (212.576–299.709)	236.321 (227.664–389.725)	0.05#	−30.103 (−88.645–0.085)	229.340 (219.533–263.887)	233.455 (212.470–281.023)	0.583#	−2.162 (−12.046–7.652)
Oleic acid	C18:1 n-9	281.487 (269.480–321.058)	340.480 (276.503–421.203)	0.06#	−31.176 (−53.726–−1.625)	318.619 (299.014–371.419)	348.144 (292.854–372.476)	0.269	0.695 (−52.065–9.262)
Linoleic acid	C18:2 n-6	**224.086** **(210.522–245.124)**	**283.474** **(208.235–314.045)**	**0.041#**	**−27.407** **(−62.497–−1.267)**	233.857 (224.404–307.236)	261.821 (220.0564–285.121)	0.48#	−3.814 (−58.30–14.185)
α-Linoleic acid	C18:3 n-3 alpha	**4.471** **(2.936–5.837)**	**4.806** **(3.553–9.541)**	**0.034#**	**−1.139** **(−1.739–−0.271)**	4.794 (3.947–8.60)	5.397 (4.010–8.294)	0.53#	−0.200 (−0.8587–0.5433)
Eicosenoic acid	C20:1 n-9	**5.771** **(4.362–7.056)**	**6.744** **(5.836–7.281)**	**0.031**	**−0.394** **(−2.049–0.093)**	5.786 (4.810–6.509)	6.068 (4.892–6.912)	0.084#	−0.558 (−0.909–0.081)
Eicosa-dienoic acid	C20:2 n-6	4.079 (3.372–4.719)	4.529 (4.208–5.115)	0.18	−0.194 (−0.890–0.052)	4.072 (3.703–4.422)	4.089 (3.970–4.395)	0.239#	−0.055 (−0.267–0.039)
Dihomo-γ-Linoleic acid	C20:3 n-6	**32.6767** **(25.9329–36.5855)**	**39.1490** **(34.5242–41.1978)**	**0.013**	**−4.2116** **(−7.1576–0.5321)**	33.3307 (29.9956–37.4817)	33.2568 (27.5007–38.7330)	0.741	0.1820 (−4.7518–2.7220)
Arachidonic acid	C20:4 n-6	**269.517** **(261.385–284.765)**	**280.996** **(265.664–368.674)**	**0.015#**	**−16.985** **(−64.073–7.210)**	298.174 (278.647–311.914)	292.519 (279.776–297.850)	0.281	15.277 (−13.874–23.519)
Eicosapenta-enoic acid	C20:5 n-3	12.140 (8.772–15.255)	13.310 (8.181–15.870)	0.738	−1.152 (−3.982–−0.3250)	13.536 (9.583–15.274)	12.409 (9.473–14.470)	0.465	−0.089 (−1.213–1.701)
Docosanoate	C22:0	**2.671** **(2.251–4.065)**	**3.149** **(2.482–6.082)**	**0.019#**	**−0.605** **(−2.533–−0.079)**	2.843 (2.554–4.023)	3.065 (2.440–3.647)	0.638#	0.046 (−0.307–0.450)
Erucic acid	C22:1 n-9	**6.346** **(6.018–7.581)**	**7.865** **(7.411–10.785)**	**0.019#**	**−1.251** **(−4.767–−0.811)**	6.286 (5.794–6.636)	6.751 (6.034–7.203)	0.308#	−0.474 (−1.131–0.283)
Docosa-pentaenoic acid ω-3	C22:5 n-3	36.456 (34.872–39.675)	40.716 (37.685–51.770)	0.084#	−2.107 (−7.691–1.232)	39.810 (37.326–43.067)	38.287 (37.40–41.891)	0.399	1.194 (−1.344–2.491)
Docosapentaenoic acid ω-6	C22:5 n-6	**4.765** **(3.293–5.372)**	**5.638** **(4.832–6.071)**	**0.012**	**−0.802** **(−1.363–−0.420)**	4.880 (3.968–5.815)	5.269 (4.577–5.580)	0.574	−0.086 (−0.339–0.283)
Docosahexa-enoic acid	C22:6 n-3	197.746 (167.709–213.221)	200.355 (173.026–240.191)	0.088	−26.823 (−42.029–−2.051)	192.054 (166.189–226.601)	190.915 (169.069–223.006)	0.59	2.531 (−10.018–9.290)
Lignocerine acid	C24:0	**5.304** **(4.366–7.143)**	**6.621** **(5.327–11.023)**	**0.032**	**−1.449** **(−3.465–0.208)**	5.391 (4.744–6.764)	5.564 (4.536–7.146)	0.239#	0.623 (−0.427–1.058)
Nervonic acid	C24:1 n-9	10.635 (8.849–12.715)	12.092 (11.196–13.128)	0.112	−0.149 (−3.933–0.275)	11.454 (9.683–13.231)	10.575 (9.739–13.225)	0.921	0.249 (−0.726–0.852)

Notes: A, arterial blood; V, venous blood. Median (IQR). A–V difference; arterio–venous difference.

**Table 3 metabolites-12-00269-t003:** Effects of hemodialysis on total fatty acids in plasma and RBCs in the CKD patients before (pre-HD) and at cessation (post-HD) of hemodialysis (*n* = 12 each).

Fatty Acids	Pre-HD Median (IQR)				Post-HD Median (IQR)			
	Pre-HD Arterial	Pre-HD Venous	*p* Value, *t* Test (# Paired Wilcoxon Test)	Pre-HD A–V Difference	Post-HD Arterial	Post-HD Venous	*p* Value, *t* Test (# Paired Wilcoxon Test)	Post-HD A–V Difference
**Total-Plasma (μg/mL)**								
Total SFA	980.036 (709.890–1195.828)	819.448 (704.758–1313.913)	0.331	41.488 (−11.696–81.741)	1056.347 (772.398–1326.228)	927.555 (680.685–1202.097)	0.091	106.329 (39.294–178.681)
Total MUFA	844.511 (668.6085–1026.8815)	781.091 (650.605–1134.145)	0.452	20.134 (−21.531–82.456)	913.802 (712.350–1110.189)	809.196 (656.101–1045.294)	0.076	68.754 (42.861–95.648)
PUFA n-3	130.480 (109.134–195.917)	122.888 (118.934–168.358)	0.433#	2.373 (−5.795–14.639)	138.296 (124.327–205.347)	118.736 (109.134–160.080)	0.117#	8.880 (2.828–20.122)
PUFA n-6	744.004 (722.153–929.411)	798.519 (686.264–895.465)	0.433#	18.340 (−38.246–78.474)	821.264 (748.646–957.803)	807.937 (676.254–859.548)	0.094	79.758 (23.246–114.425)
Total-PUFA	857.990 (820.586–1136.295)	908.785 (782.665–1076.363)	0.433#	22.907 (−43.998–96.703)	947.742 (872.870–1160.028)	915.718 (784.884–1004.691)	0.103	89.139 (25.349–134.547)
**Total-RBC (μg/g)**								
Total SFA	**595.703** **(513.040–757.739)**	**670.018** **(553.384–972.688)**	**0.027**	**−56.325** **(−277.577–2.217)**	624.941 (551.351–746.70)	591.402 (534.765–831.952)	0.182#	−28.462 (−94.268–36.130)
Total MUFA	**322.825** **(305.306–380.602)**	**387.006** **(329.542–476.590)**	**0.034#**	**−46.965** **(−63.638–9.095)**	376.076 (332.646–409.804)	394.564 (330.508–432.247)	0.188	−5.169 (−62.982–7.602)
PUFA n-3	257.166 (218.402–273.419)	261.911 (235.961–312.850)	0.078	−37.862 (−48.270–4.691)	246.884 (223.435–303.050)	246.460 (218.649–284.657)	0.579	4.399 (−11.394–14.169)
PUFA n-6	**538.579** **(527.422–560.087)**	**625.916** **(544.230–707.882)**	**0.008#**	**−46.288** **(−97.802–19.074)**	588.363 (554.219–649.017)	605.591 (539.772–637.355)	0.814#	15.263 (−80.730–38.335)
Total-PUFA	**784.818** **(758.405–818.550)**	**905.273** **(795.178–1041.732)**	**0.023#**	**−78.630** **(−162.449–37.522)**	876.443 (778.130–892.759)	843.795 (801.30–888.677)	0.937#	33.365 (−92.123–67.535)

Notes: A, arterial blood; V, venous blood. Median (IQR). A–V difference; arterio–venous difference.

**Table 4 metabolites-12-00269-t004:** Effect of hemodialysis on polyunsaturated fatty acid ratios in plasma and RBCs in the CKD patients before (pre-HD) and at cessation (post-HD) of hemodialysis (*n* = 12 each).

Ratio	Pre-HD Median (IQR)			Post-HD Median (IQR)		
	Pre-HD Arterial	Pre-HD Venous	*p* Value, *t* Test (# Paired Wilcoxon Test)	Post-HD Arterial	Post-HD Venous	*p* Value, *t* Test (# Paired Wilcoxon Test)
**Total-Plasma**						
DHA + EPA/AA	0.751 (0.627–0.875)	0.751 (0.627–0.875)	0.573	0.721 (0.610–0.873)	0.652 (0.595–0.817)	0.358
EPA/AA	0.105 (0.067–0.154)	0.098 (0.070–0.154)	0.3	0.101 (0.069–0.152)	0.081 (0.072–0.130)	0.61#
DHA/AA	0.620 (0.564–0.702)	0.636 (0.559–0.679)	0.754#	0.609 (0.534–0.695)	0.578 (0.512–0.661)	0.47
DHA/EPA	6.625 (4.853–7.133)	6.554 (5.180–7.437)	0.608	6.533 (5.242–7.165)	6.589 (5.856–7.350)	0.179
n-3/n-6	0.183 (0.151–0.212)	0.183 (0.151–0.212)	0.878	0.186 (0.151–0.210)	0.165 (0.154–0.218)	0.84
**Total-RBC**						
Omega-3 quotient	11.072 (9.634–13.067)	11.375 (7.710–12.171)	0.776	11.377 (9.737–12.382)	11.374 (8.304–12.544)	0.217
DHA + EPA/AA	0.759 (0.598–0.823)	0.655 (0.521–0.904)	0.835	0.738 (0.567–0.805)	0.703 (0.615–0.811)	0.851
EPA/AA	0.042 (0.030–0.058)	0.034 (0.030–0.058)	0.445	0.043 (0.032–0.055)	0.042 (0.034–0.051)	0.575
DHA/AA	0.710 (0.570–0.769)	0.620 (0.505–0.847)	0.908	0.695 (0.526–0.737)	0.661 (0.583–0.749)	0.759
DHA/EPA	16.161 (12.819–18.593)	15.261 (14.596–21.717)	0.814#	16.428 (12.26–18.764)	17.380 (13.466–20.360)	0.929
n-3/n-6	0.443 (0.404–0.493)	0.410 (0.352–0.505)	0.822	0.445 (0.361–0.485)	0.410 (0.388–0.462)	0.356

Notes: A, arterial blood; V, venous blood. Median (IQR).

**Table 5 metabolites-12-00269-t005:** Effect of hemodialysis on the ratio of desaturase and peroxisome function in plasma and RBC in the CKD patients before (Pre-HD) and at cessation (Post-HD) of hemodialysis (*n* = 12 each).

Ratio	Pre-HD Median (IQR)			Post-HD Median (IQR)		
	Pre-HD Arterial	Pre-HD Venous	*p* Value, *t* Test (# Paired Wilcoxon Test)	Post-HD Arterial	Post-HD Venous	*p* Value, *t* Test (# Paired Wilcoxon Test)
**Total-Plasma**						
DHA/DPA	9.146 (6.889–11.009)	8.987 (7.0171–10.7715)	0.208	8.779 (6.518–11.679)	9.047 (6.545–13.840)	0.281
C20:4 n-6/C20:3 n-6 (Δ5 SCD)	3.663 (3.345–4.293)	3.573 (3.295–4.030)	0.201	3.799 (3.334–4.312)	3.906 (3.315–4.759)	0.849
C18:3 n-6/C18:2 n-6 (Δ6 SCD)	0.016 (0.010–0.019)	0.015 (0.010–0.019)	0.456#	0.017 (0.013–0.020)	0.017 (0.011–0.020)	0.223#
C16:1 n-7/C16:0 (Δ9 SCD)	0.095 (0.079–0.100)	0.088 (0.081–0.102)	0.79	0.084 (0.072–0.094)	0.086 (0.076–0.092)	0.314
C18:1 n-9/C18:0 (Δ9 SCD)	5.807 (5.315–6.423)	5.764 (4.642–7.018)	0.937#	6.290 (4.973–6.424)	6.005 (4.969–6.274)	0.484
**Total-RBC**						
DHA/DPA	5.283 (3.609–5.794)	5.015 (3.541–5.929)	0.813	5.110 (3.990–5.553)	4.985 (4.368–5.839)	0.859
C20:4 n-6/C20:3 n-6 (Δ5 SCD)	9.405 (7.575–10.389)	8.057 (6.667–10.316)	0.638#	8.659 (7.893–10.517)	8.145 (7.509–10.891)	0.44
C18:3 n-6/C18:2 n-6 (Δ6 SCD)	0.011 (0.006–0.013)	0.011 (0.007–0.013)	0.651	0.011 (0.008–0.013)	0.011 (0.007–0.014)	0.87
C16:1 n-7/C16:0 (Δ9 SCD)	0.042 (0.035–0.058)	0.044 (0.033–0.057)	0.903	0.043 (0.039–0.052)	0.048 (0.038–0.057)	0.143
C18:1 n-9/C18:0 (Δ9 SCD)	1.262 (1.068–1.500)	1.258 (0.914–1.523)	0.833	1.320 (1.184–1.593)	1.338 (1.177–1.484)	0.554

Notes: A, arterial blood; V, venous blood. Median (IQR).

## Data Availability

Data is contained within the article or Appendix A.

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
