# Peer review of "Bioaccumulation of Blood Long-Chain Fatty Acids during Hemodialysis"

_metabolites, 2022, doi:10.3390/metabo12030269_

Round 1

Reviewer 1 Report

Although the research is interesting I struggled to understand some of the interpretations ie. "we observed that AV differences in LFCA were affected by HD treatment in RBC?"  Do you mean There were differences in arterial and venous LFCA within RBCs after HD?   The next sentence in the abstract is also very confusing -  "Numerous SFAs, MLFAs and PLFAs showed negative AV differences in LCFA were affected by HD treatment in RBCs before HD." Please clarify this statement.  What do you mean by negative AV difference. These interpretations are difficult to understand and take away from the research findings.  HD reduced these differences, again what differences? 

Author Response

Reviewer 1:

Comments to the Author

      Thank you very much for your positive comments and constructive criticism.

  1. Although the research is interesting I struggled to understand some of the interpretations ie. "we observed that AV differences in LFCA were affected by HD treatment in RBC?"  Do you mean There were differences in arterial and venous LFCA within RBCs after HD?   

Our response:

We apologize for the imprecision. Our study is the first to study the A-V differences of blood FAs content between the arterial blood and the venous blood.  Text has been changed according to your comment (abstract, introduction section, figure legend). We now also better explain the meaning of negative differences in arterial and venous FAs. Please note that this approach has been used by others to study ammonia levels, glucose metabolism and NO formation (cf. introduction section and references). We applied this approach to better understand biotransformation/bioaccumulation of FAs in vivo, particularly whether or not the peripheral tissues, especially the muscles in the upper limbs, either produce, store or degrade part of the metabolites that pass through them in response to dialysis treatment. Our data demonstrate that A-V differences in fatty acids status of LCFA are present and active in mature erythrocytes and their bioaccumulation is sensitive to single HD treatment. Our data (pre-HD) is consistent with bioaccumulation of numerous FAs in RBCs.

  1. The next sentence in the abstract is also very confusing -  "Numerous SFAs, MLFAs and PLFAs showed negative AV differences in LCFA were affected by HD treatment in RBCs before HD." Please clarify this statement. 

Our response:  We apologize. Text has been corrected.

What do you mean by negative AV difference. These interpretations are difficult to understand and take away from the research findings.  HD reduced these differences, again what differences? 

Our response:  Please, see our comments above.

Reviewer 2 Report

General comments: The title is misleading, only a small part of the article/results are about LCFA. In the second paragraph of the introduction PUFA, MUFA, and SFA are mentioned but not a word about LCFA. Are there any data on LCFA and CVD risk, correlation between dietary supplementation of FA and LCFA level?

Are there any data on peritoneal dialysis and FAs level?

The result tables are huge, often spread over several pages, which makes them difficult to read. Perhaps it will be enough to mark only what was  statistically significant without giving p-values - it will reduce the amount of data by two columns. Is it really necessary to use values to 4 decimal places?

specific comments:

-line 44: "ell-membrane" - cell?

- no explanation of the use of colors in the description of figure 1

- section 2.2 - please explain in brief

Author Response

Reviewer 2:

              Thank you for your constructive comments.

  1. General comments: The title is misleading, only a small part of the article/results are about LCFA. In the second paragraph of the introduction PUFA, MUFA, and SFA are mentioned but not a word about LCFA. Are there any data on LCFA and CVD risk, correlation between dietary supplementation of FA and LCFA level?

Our response: Thank you. Title has been changed. LCFA refers to long-chain fatty acids with more than 12 carbon atoms (C12). SFA, MUFA, and PUFA are now clearly explained in the text and the new figure 1.

  1. Are there any data on peritoneal dialysis and FAs level?

Our response:  To our knowledge, our study is first to study the A-V differences of blood FAs content between the arterial blood and the venous blood. We applied this approach to better understand bioaccumulation and biotransformation of FAs in vivo, particularly whether or not the peripheral tissues, especially the muscles in the upper limbs, either produce, store or degrade part of the FAs that pass through them in response to dialysis treatment. We were particularly interested to understand how the FAs in RBC and plasma are modified by hemodialysis treatment, which is known to cause oxidative stress, RBC-endothelial interactions, vascular damage and inflammation.

              To our knowledge, there are no studies on A-V differences of blood FAs content in peritoneal patients. Nevertheless, we found that some researchers have studied blood levels of FAs in peritoneal dialysis patients and have come to similar conclusions as we have, namely that MUFA and SFA are likely affected by dialysis treatment. Your suggestion gives us a good direction for our future studies.

An W., Kim S., Kim K., Lee S., Park Y., Kim H., Vaziri N. Comparison of fatty acid contents of erythrocyte membrane in hemodialysis and peritoneal dialysis patients. J. Ren. Nutr. 2009;19:267–274. doi: 10.1053/j.jrn.2009.01.027

Yerlikaya F.H., Mehmetoglu I., Kurban S., Tonbul Z. Plasma fatty acid composition in continuous ambulatory peritoneal dialysis patients: An increased omega-6/omega-3 ratio and deficiency of essential fatty acids. Ren. Fail. 2011;33:819–823. doi: 10.3109/0886022X.2011.601831.

Pazda M., Stepnowski P., Sledzinski T., Chmielewski M., Mika A. Suitability of selected chromatographic columns for analysis of fatty acids in dialyzed patients. Biomed. Chromatogr. 2017;31:e4006. doi: 10.1002/bmc.4006.

An W., Lee S., Son Y., Kim S., Kim K., Han J., Bae H., Park Y. Effect of omega-3 fatty acids on the modification of erythrocyte membrane fatty acid content including oleic acid in peritoneal dialysis patients. Prostaglandins Leukot. Essent. Fatty Acids. 2012;86:29–34. doi: 10.1016/j.plefa.2011.10.009. 

Ban-Hock Khor, Sreelakshmi Sankara Narayanan, Karuthan Chinna, Abdul Halim Abdul Gafor, Zulfitri Azuan Mat Daud, Pramod Khosla, Kalyana Sundram, Tilakavati Karupaiah. Blood Fatty Acid Status and Clinical Outcomes in Dialysis Patients: A Systematic Review. Nutrients. 2018 Oct; 10(10): 1353. doi: 10.3390/nu10101353

  1. The result tables are huge, often spread over several pages, which makes them difficult to read. Perhaps it will be enough to mark only what was statistically significant without giving p-values - it will reduce the amount of data by two columns. Is it really necessary to use values to 4 decimal places?

Our response: We agree. In our revised manuscript, the decimal places were shortened and rounded up and down accordingly. We understand that the tables are extensive, but we believe that reporting p-values is important to help readers understand the content of our work.

Specific comments:

-line 44: "ell-membrane" - cell?

Our response: Corrected.

- no explanation of the use of colors in the description of figure 1

Our response:  Figure legend has been changed.

- section 2.3 - please explain in brief

Our response: Methods section has been extended. Please, see the revised version of our manuscript.
